# Light-Triggered Cellular Delivery of Oligonucleotides

**DOI:** 10.3390/pharmaceutics11020090

**Published:** 2019-02-21

**Authors:** Leena-Stiina Kontturi, Joep van den Dikkenberg, Arto Urtti, Wim E. Hennink, Enrico Mastrobattista

**Affiliations:** 1Department of Pharmaceutics, Utrecht Institute for Pharmaceutical Sciences (UIPS), Utrecht University, 3584 CG Utrecht, The Netherlands; leena.kontturi@helsinki.fi (L.-S.K.); J.B.vandenDikkenberg@uu.nl (J.v.d.D.); w.e.hennink@uu.nl (W.E.H.); 2Drug Research Program, Division of Pharmaceutical Biosciences, Faculty of Pharmacy, University of Helsinki, 00790 Helsinki, Finland; arto.urtti@helsinki.fi; 3School of Pharmacy, Faculty of Health Sciences, University of Eastern Finland, 70211 Kuopio, Finland; 4Institute of Chemistry, St Petersburg State University, Petergoff, 198540 St Petersburg, Russia

**Keywords:** oligonucleotide delivery, light-activated release, intracellular release, liposome, indocyanine green

## Abstract

The major challenge in the therapeutic applicability of oligonucleotide-based drugs is the development of efficient and safe delivery systems. The carriers should be non-toxic and stable in vivo, but interact with the target cells and release the loaded oligonucleotides intracellularly. We approached this challenge by developing a light-triggered liposomal delivery system for oligonucleotides based on a non-cationic and thermosensitive liposome with indocyanine green (ICG) as photosensitizer. The liposomes had efficient release properties, as 90% of the encapsulated oligonucleotides were released after 1-minute light exposure. Cell studies using an enhanced green fluorescent protein (EGFP)-based splicing assay with HeLa cells showed light-activated transfection with up to 70%–80% efficacy. Moreover, free ICG and oligonucleotides in solution transfected cells upon light induction with similar efficacy as the liposomal system. The light-triggered delivery induced moderate cytotoxicity (25%–35% reduction in cell viability) 1–2 days after transfection, but the cell growth returned to control levels in 4 days. In conclusion, the ICG-based light-triggered delivery is a promising method for oligonucleotides, and it can be used as a platform for further optimization and development.

## 1. Introduction

Therapeutics based on oligonucleotides have significant potential for the treatment of a wide variety of diseases [1,2]. In principle, any disease with a known genetic origin can be treated by modifying genetic functions with oligonucleotide-based drugs. Compared to traditional pharmaceuticals, this approach has several advantages, including specificity, potency, and possibility for a rapid and rational drug design. The major limitation for the clinical translation of these therapeutics is the difficulty of in vivo delivery; as large, anionic macromolecules that are prone to degradation by nucleases, oligonucleotides require sophisticated carrier systems to enable delivery into the target cells [3,4]. An optimal carrier protects oligonucleotides from enzymatic degradation and clearance, and transfers them selectively into the cytoplasm of the target cells with minimal toxicity.

The most investigated synthetic vectors for oligonucleotides are lipid-based nanoparticles [5,6,7]. Commonly, these carriers contain cationic lipids that enable high loading capacity by complexing with negatively charged oligonucleotides and efficient intracellular delivery by interacting with the negatively charged cell membranes. However, the utility of cationic liposomes in vivo is limited, as excess positive charge results in toxicity, innate immune activation, and poor pharmacokinetic properties [8,9,10]. Neutral or anionic liposomes show less interaction with serum proteins and complement components and, consequently, are less toxic and have better pharmacokinetic profiles. Yet, as non-cationic lipids do not interact with cellular membranes as efficiently as cationic ones, cellular uptake and intracellular release of entrapped oligonucleotide cargo with neutral liposomes is usually poor. Currently, the most promising lipid formulations contain ionizable lipids [11,12,13]. These lipids switch charge pH-dependently, enabling neutral particles at physiological pH in the blood circulation and in the extracellular space of tissues, and positive charge in the acidic environment of endosomes after cellular internalization.

In general, liposomal drug delivery is associated with poorly controlled and insufficient cytosolic oligonucleotide release. To improve the control and effectiveness of cytosolic delivery and drug release at the target site, systems that are activated by external or internal signals, such as temperature, pH, ultrasound, specific enzymes, magnetic field and light, have been developed [14,15]. In the present study, we applied a previously developed light-triggered liposomal system for oligonucleotide delivery [16,17,18]. The system consists of thermosensitive liposomes with indocyanine green (ICG) as the photosensitizing agent. The light sensitivity is based on the photothermal ability of ICG to absorb light energy and convert it to heat [18,19]; when the temperature-sensitive liposomes containing ICG are exposed to light, the released heat creates a localized temperature increase, leading to fluidization of the thermosensitive lipid membranes and release of the encapsulated drug.

ICG injections have been approved by the US Food and Drug Administration (FDA) and European Medicines Agency (EMA) for fluorescence-based clinical imaging [20,21]. Compared to the most commonly used photothermal agents, gold and carbon nanomaterials, ICG has certain advantages: (1) ICG has absorption maximum at the near infrared (NIR) range, enabling excitation at a safe wavelength of 800 nm that penetrates into tissues [22,23]. (2) As an organic molecule, ICG can be conveniently incorporated into delivery systems. Also, processes for particle size control, such as extrusion and microfluidization, can be used, as the presence of ICG does not limit the size of the carrier [17]. (3) Since ICG is a fluorescent compound, it enables imaging-guided drug delivery in certain tissues [24,25,26]. (4) The safety profile of ICG is well-documented, while the long-term toxicity of non-biodegradable inorganic nanoparticles is unknown and they have not been approved for clinical use [27,28].

We have previously shown that the ICG-containing liposomes are functional in light-triggered release of small and large fluorescently labeled model compounds [16]. In the present work, we extended the concept to the delivery of oligonucleotides. Our aim was to investigate the effects of light induction and ICG on cellular delivery of oligonucleotides and liposomal oligonucleotides.

## 2. Materials and Methods

### 2.1. Materials

1,2-Dipalmitoyl-sn-glycero-3-phosphocholine (DPPC), 1,2-distearoyl-sn-glycero-3-phosphocholine (DSPC) and 1,2-distearoyl-sn-glycero-3-phosphoethanolamine-*N*-[methoxy(polyethylene glycol)-2000] (DSPE-PEG) were bought from Lipoid (Ludwigshafen, Germany). 1-stearoyl-2-hydroxy-sn-glycero-3-phosphocholine (Lyso PC) was from Avanti Polar Lipids, Inc. (Alabaster, AL, USA). The oligonucleotides used in this study were a splice switching antisense oligonucleotide (SSO) restoring correct splicing of EGFP [29] and an siRNA against luciferase. The sequences of the oligonucleotides are the following:
SSO: 5′-GCT ATT ACC TTA ACC CAG-3′siRNA: sense 5′-CUUACGCUGAGUACUUCGAdTdT-3′anti-sense 5′-UCGAAGUACUCAGCGUAAGdTdT-3′
Underlined bases indicate a 2′-*O*-methyl modification. dT indicates deoxyribonucleic acid bases with phosphorothioate (PS) bonds. The SSO consists completely of PS bonds. The SSO was purchased from Biosearch Technologies (Petaluma, CA, USA) and the siRNA from Integrated DNA technologies (Leuven, Belgium). Cell medium and supplements were from GibcoBRL, Thermo Fisher Scientific (Naarden, The Netherlands). Indocyanine green purchased from Sigma-Aldrich (St. Louis, MO, USA) was the United States Pharmacopeia (USP) Reference Standard (mw. 775 g/mol). All other compounds were bought from Sigma-Aldrich (St. Louis, MO, USA) unless otherwise mentioned. Fluorescence measurements of ICG and the Ribogreen assay were performed with a Jasco FP8300 Spectrofluorometer with micro-well plate reader (JASCO Benelux BV., De Meern, The Netherlands). An 808N10W laser system with a circular beam of 7 mm in diameter was used for the light triggering studies (Changchun Dragon Lasers Co., Ltd., Changchun, China). The output light intensities with different power settings were measured using a P-9710-1 optometer with RCH-102-2 custom-made detector head (Te Lintelo Systems BV, Zevenaar, The Netherlands). Light intensities (mW/cm^2^) corresponding to the power settings of 1–10 W are shown in the Appendix A.

### 2.2. Liposome Preparation

Liposomes were prepared by a lipid film hydration method using a composition of DPPC/DSPC/Lyso PC/DSPE-PEG at a molar ratio of 75:15:10:4, respectively. The lipids dissolved in chloroform were mixed and dried to a thin lipid film by rotary evaporation. Residual chloroform was removed under a nitrogen flow for 30 min. The film was hydrated at 55 °C for 1 h with 2.5 mg/mL oligonucleotides dissolved in 20 mM HEPES buffer, pH 7.4. To promote encapsulation of oligonucleotides, a high lipid concentration of 100 mM (typically, 50 µmol of total lipids and 0.5 mL of hydration solution) was used. The formed liposomes were extruded 5 times through a track-etched polycarbonate membrane (Whatman Nuclepore, GE Healthcare, Chicago, IL, USA) with 100-nm pore size using a syringe extrusion device (Avanti Polar Lipids Inc., Alabaster, AL, USA). The free, unencapsulated oligonucleotides were separated by ultracentrifugation for 1 h at 55,000 rpm at 4 °C for 2–3 times, and the liposomes were dispersed in 20 mM HEPES, 140 mM NaCl, pH 7.4.

For ICG incorporation, the liposomes were incubated in ICG solution of 1 mg/mL (in 20 mM HEPES, 140 mM NaCl, pH 7.4) for 1 h in rotation at room temperature. The volumes of the incubated liposomes and ICG solution were adjusted to molar ratios in the range of 1/25–1/200 ICG to lipid. The amount of ICG in the liposomes was determined by separating the free ICG by ultracentrifugation (1 h at 55,000 rpm at 4 °C) after incubation. The amount of free ICG in the supernatant was determined based on ICG fluorescence that was measured at 770/810 nm (excitation/emission wavelengths), and the concentration was determined using a calibration curve. As the percentage of non-incorporated ICG was less than 10% even at the highest ICG concentration, the separation step by ultracentrifugation was not considered necessary and was not carried out in further experiments. The addition of ICG to the liposomes was done always immediately before the experiments.

### 2.3. Liposome Characterization

#### 2.3.1. Size

The mean particle size and polydispersity index (PDI) were measured by dynamic light scattering (DLS) with a Malvern CGS-3 multiangle goniometer with He–Ne laser source (λ = 632.8 nm, 22 mW output power) using an angle of 90° (Malvern Instruments, Malvern, UK). For the measurement, liposome samples were diluted to 0.25 mM (lipid concentration) in 20 mM HEPES with 140 mM NaCl, pH 7.4.

#### 2.3.2. Zeta-Potential

The zeta-potential of the liposomes was measured on a Zetasizer Nano-Z (Malvern Instruments) with samples diluted to 0.25 mM (lipid concentration) in 20 mM HEPES, pH 7.4.

#### 2.3.3. Phase Transition Temperature (*T_m_*)

Differential scanning calorimetry (DSC) (Discovery DSC, TA instruments, New Castle, DE, USA) was used to determine the *T_m_* values of the liposomes. Liposome sample with a concentration of 50 mM and a reference sample (20 mM HEPES with 140 mM NaCl, pH 7.4) were placed in hermetically sealed aluminum pans, and heated from 20 to 60 °C at a rate of 0.5 °C/min. *T_m_* represents the peak temperature of the endotherm recorded during the heating scan.

#### 2.3.4. Encapsulation Efficiency

The amount of SSO or siRNA encapsulated inside the liposomes was determined using Quant-iT™ RiboGreen® RNA Assay Kit (Thermo Fischer Scientific, Waltham, MA, USA) according to the manufacturer’s protocol. The measurement was performed for non-treated liposomes and for liposomes disrupted with 0.5% Triton-X 100. As the Ribogreen reagent does not penetrate liposomal membranes, the signal measured after treatment with Triton-X 100 represents the oligonucleotide concentration entrapped inside the liposomes. Concentrations of the samples were calculated based on calibration curves prepared with (1) oligonucleotides in the presence of empty liposomes and (2) oligonucleotides in the presence of empty liposomes and Triton-X 100. The encapsulation efficiency was calculated using the formula: % encapsulation = (ON_t_ − ON_0_)/ON_i_ × 100, where ON_t_ = oligonucleotide concentration of Triton-X 100 treated liposomes, ON_0_ = oligonucleotide concentration of non-treated liposomes, and ON_i_ = initial oligonucleotide concentration. Measurements were done at wavelengths of 480/520 nm.

#### 2.3.5. Light-Induced Oligonucleotide Release

The light-induced release of SSO or siRNA from the liposomes was determined by measuring the concentrations of non-treated liposomes, liposomes after light exposure, and liposomes treated with 0.5% Triton-X 100. The liposomes were diluted to 1 mM (lipid concentration) in buffer, heated to 37 °C on a thermomixer heating device and exposed to 808 nm light with the intensity of 370 mW/cm^2^ for 1 min. Control samples were kept at similar conditions, but were shielded from the light. Non-treated samples at +4 °C represented the background signal and Triton-X 100 treated samples were set at 100% release. The oligonucleotide concentrations were measured immediately after the light exposure using Quant-iT RiboGreen RNA Assay Kit as described in the previous section, and the release percentage was calculated using the formula: % released = (ON_l_ − ON_0_)/(ON_t_ − ON_0_) × 100, where ON_l_ = oligonucleotide concentration of light exposed liposomes, ON_0_ = oligonucleotide concentration of non-treated liposomes, and ON_t_ = oligonucleotide concentration of Triton-X 100 treated liposomes.

Both the SSO- and siRNA-encapsulated ICG liposomes were prepared and characterized as described above, while cell studies were performed using only the SSO-encapsulated liposomes. The purpose of the siRNA-encapsulated liposomes was to investigate if the light-triggered release of siRNA (*M_W_* ~ 14 kDa) differed from the release of SSO (*M_W_* ~ 7 kDa).

### 2.4. Cell Studies

#### 2.4.1. Cell Line

To study the ability of the light-activated liposomes to deliver oligonucleotides intracellularly, an EGFP-based splicing assay with HeLa S3 cells was used [29]. The assay was based on a construct where a C-to-T mutation at nucleotide 654 of the human β-globin intron-2 was inserted in the EGFP cDNA (IVS2-654), preventing correct translation of EGFP. Delivery of SSO (antisense oligonucleotide directed to position 654) blocked the aberrant splice site and restored the correct splicing of the EGFP precursor mRNA, generating properly translated EGFP. In this approach, antisense activity of SSO was directly proportional to up-regulation of EGFP in cells transfected with the IVS2-654 EGFP construct, thus providing a positive, quantitative readout.

#### 2.4.2. Cell Culture

HeLa S3 IVS2-654 EGFP cells were cultured in Dulbecco’s Modified Eagle’s Medium with high glucose supplemented with 10% (*v*/*v*) fetal bovine serum and 400 μg/mL G418 at 37 °C under a humidified atmosphere containing 5% CO_2_. Cells were routinely passaged twice a week and used for the experiments at passages 5–20. Cells were regularly tested for mycoplasma and found to be negative.

#### 2.4.3. Transfection Studies

HeLa S3 IVS2-654 EGFP cells were seeded on white μView clear-bottom 96-well plates (Greiner Bio-One B.V., Alphen aan de Rijn, The Netherlands) at a density of 9000 cells/well. After attachment (5–6 h after seeding), liposomes diluted in growth medium were added to the cells for overnight incubation. Next day, the liposomes were removed, the cells were washed with phosphate buffered saline, and the light triggering was performed. In the light triggering set-up, the cell culture plate was placed on a thermomixer heater to keep the temperature at 37 °C and the cells were exposed to 808 nm light, while control samples on the same plate were shielded from the light. After light exposure, the cells were transferred back to the culture incubator (37 °C under a humidified atmosphere containing 5% CO_2_). Transfection was typically measured either 24 or 48 h after light exposure (in some experiments, also the time points of 72 and 96 h were used). Detection was done by confocal imaging using a Cell Voyager CV7000s high-content confocal imager (Yokogawa, Tokyo, Japan). The nuclei were stained by incubating the cells in 2 μg/mL Hoechst 33342 (Life Technologies, Bleiswijk, The Netherlands) for 15 min at 37 °C. Imaging was done by acquiring 20–30 images with 3–4 z-stacks from each well using a 20× objective.

The effect of the following variables on transfection efficacy was investigated: (1) liposome concentration, (2) ICG concentration of the liposomes, (3) intensity of light, and (4) duration of light exposure. In addition, possible changes in transfection efficacy were followed for 48, 72, and 96 h after transfection and light triggering. Finally, the ability of free ICG or empty ICG liposomes in combination with free SSO to induce transfection upon light triggering was studied. Lipofectamine® 3000 (Thermo Fisher Scientific, Waltham, MA, USA) was used as a positive control according to the manufacturer’s protocol. As a negative control, an antisense oligonucleotide directed to position 705 of the cDNA was used [30]. This is an oligonucleotide of the same length as the active SSO, but complementary to a different position and will thus not result in altered splicing.

#### 2.4.4. Cytotoxicity

Cytotoxicity of light exposure and/or the liposome formulation were measured by calculating Hoechs-labeled nuclei (2 μg/mL Hoechst 33342 for 15 min at 37 °C) from confocal images. Numbers of the treated cells were compared to numbers of cells without any treatment and reported as percentage values (cells without treatment set as 100%). Effect of light intensity (370–1500 mW/cm^2^), liposome concentration (0.5–1.4 mM, overnight incubation), and the combination of these were studied, and the cell numbers were measured at 48, 72, and 96 h after treatment.

#### 2.4.5. Image Analysis

Customized image analysis protocols were developed with Columbus Software (version 2.7.1; PerkinElmer Inc., Waltham, MA, USA). The determination of transfection efficacy was based on the intensity of EGFP fluorescence signal in cell cytoplasm. The percentage of transfected cells was obtained by setting a threshold value for the EGFP intensity and by separating the cell population using this value into transfected (intensity above the threshold) and non-transfected (intensity below the threshold) cells. The threshold was set to give less than 1 as the transfection percentage (<1% transfection efficacy) for the non-treated control cells. Details of the analysis can be found from Appendix A. Cell numbers were determined by calculating Hoechst-labeled nuclei.

## 3. Results

### 3.1. Characterization of the Oligonucleotide-Encapsulated ICG Liposomes

The mean diameter of the liposomes was ~150 nm with a PDI < 0.15. The incorporation of ICG did not significantly affect the liposome size. Zeta-potential values of the liposomes without ICG were negative (~ −8 mW), and inclusion of ICG further lowered the zeta-potentials. The *T_m_* values were 42.6–42.7 °C, and the incorporation of ICG lowered these values by 0.4–0.8 °C. Encapsulation efficiency of oligonucleotides was 5%–8%. Light-induced release of oligonucleotides was effective; approximately 90% of the SSO and siRNA were released after exposure to 808 nm light with the intensity of 370 mW/cm^2^ for 1 min. Results on the characterization of the liposomes are shown in Table 1; Table 2 and in Figure 1. In general, there were no significant differences in the characteristics between the liposomes encapsulated with the SSO or siRNA or empty liposomes.

### 3.2. Transfection Experiments

We studied the effects of liposomal formulation (ICG and lipid concentrations) and light exposure (intensity, duration) on transfection efficacy. Moreover, transfection efficacy at different time points was measured. Successful transfection was seen as increased EGFP fluorescence in the cell cytoplasm (Figure 2). Treatment with ICG liposomes encapsulated with the negative control oligonucleotide induced no detectable transfection (Appendix A). Light triggering was shown not to affect the activity of the SSO by comparing transfection efficacy of Lipofectamin-transfected cells with and without light exposure (Appendix A).

#### 3.2.1. Liposome Concentration

Liposome concentration had a clear effect on transfection efficacy (Figure 2). Increasing liposome concentrations led to higher transfection percentages: when the concentration was approximately tripled (from 0.5 mM to 1.4 mM), the transfection percentage increased roughly 3-fold (from 20% to 60% and from 30% to 80%) (Figure 3A,B). An increased transfection was detected in the cells treated with liposomes and exposed to light, as well as in the cells that were incubated with liposomes without light exposure. This was especially evident for liposomes having a higher concentration of ICG; with 1.4 mM liposomes of 1/25 ICG-to-lipid ratio, 14% of the cells were transfected without light exposure (Figure 3B). Yet, the transfection efficacy was considerably higher (78%) with light exposure than without light. In general, the transfection efficacies obtained with the light-triggered liposomes were in the same range as with Lipofectamine® 3000 that was used as a positive control. However, compared to Lipofectamine, the results with the light-triggered delivery were much more consistent (Appendix A).

#### 3.2.2. ICG Concentration

Increasing the ICG concentration that was added to the liposomes led to an increased transfection efficacy. The higher the ICG concentration of the liposomes, the higher the percentage of transfected cells (Figure 3C,D). Instead, cells treated with liposomes having different lipid concentrations but same ICG concentration had equal transfection percentages (Figure 3E,F). As a conclusion, efficacy of the transfection process is mainly dependent on the ICG concentration, and both lipid and SSO concentrations have a smaller impact. Moreover, the liposomes with a higher ICG concentration induced transfection in the absence of light as well. At the highest ICG concentration, about 20% of the cells were EGFP positive without light exposure (Figure 3D,F).

#### 3.2.3. Illumination Time

In general, the transfection efficacy was not affected when the illumination time was increased to longer than 1 min (Figure 4A,B). Effects of illumination times shorter than 1 min were dependent on other variables, the major determinant being ICG concentration. At a higher ICG concentration, changes in transfection efficacy with illumination times of 15 s, 30 s, and 1 min were more prominent than those observed for lower ICG concentrations. In addition, at very high ICG concentrations, some increase in transfection efficacy could be seen between 1 min and 2 min light exposures (data not shown). Consequently, the illumination time of 2 min was chosen for further studies.

#### 3.2.4. Light Intensity

The intensity of 808 nm light used to trigger the liposomes did not affect the transfection efficacy in the range of 370–1500 mW/cm^2^ (Figure 4C,D). The laser instrument used in this study limited the intensities to 370 mW/cm^2^ and higher levels. Intensities higher than 1500 mW/cm^2^ led to variability in results between repeats, and in general did not significantly improve the transfection efficacy (data not shown). As a conclusion, light intensities in the range of 370–1500 mW/cm^2^ were suitable for induction of the oligonucleotide release.

#### 3.2.5. Free ICG and SSO

During the studies, we observed that ICG and SSO can transfect cells upon light triggering without incorporation into liposomes. Therefore, experiments with free ICG or empty ICG liposomes and free SSO were performed. The transfection efficacies of free ICG and SSO, and SSO-encapsulated ICG liposomes were equal (Figure 5). In addition, a mixture of empty ICG liposomes and free-SSO transfected cells, but less effectively compared to free ICG and SSO or the SSO-encapsulated ICG liposomes (transfection efficacies of 19% versus 40% with 14 μM ICG; 45 versus 60% with 35 μM ICG, respectively). In the absence of ICG, treatment with SSO did not result in detectable levels of transfected cells.

#### 3.2.6. Time after Transfection and Light Triggering

At higher liposome concentrations of 0.7 and 1.4 mM, transfection efficacies increased from 48 to 96 h after light triggering: transfection percentages increased from ~35% to 50% with 0.7 mM liposomes and from ~55% to 70% with 1.4 mM liposomes (Figure 6). At the lower liposome concentrations of 0.25 and 0.5 mM, differences between the time points were smaller and a slight increase in the transfection percentages could be seen only between 48 and 72 h (from 10% to 14% with 0.25 mM liposomes; from 22% to 27% with 0.5 mM liposomes). After 4 days, the HeLa cells grew over-confluent making quantitative analysis of the results impossible.

### 3.3. Cytotoxicity

The light exposure or liposome incubation separately showed no toxic effects (Figure 7A,B). Slightly lower cell numbers were seen after incubation with 1.0 and 1.4 mM liposomes: cell numbers were 92%–96% of the control cell numbers at the first time point, 48 h after liposome incubation. At 72 and 96 h, cell numbers were similar in all the groups. The combination of liposomes and light exposure resulted in moderate toxic effects (Figure 7C). Especially, 48 h after light triggering, the cell numbers were lower than in the control group: 75% in cells treated with 0.5 and 0.7 mM liposomes and 65% in cells treated with 1.0 and 1.4 mM liposomes. However, cell numbers in the treated groups had increased at 72 h after light triggering and at 96 h, the cell numbers did not differ from the non-treated control group.

## 4. Discussion

The light-triggered liposome formulation used in this work is based on our previous studies on ICG liposomes [16,17], but ICG was now incorporated into the liposome dispersion. This approach avoided any interference of ICG on oligonucleotide encapsulation. As an amphiphilic compound, ICG can associate with the hydrophilic surface layer of liposomes or penetrate the hydrophobic lipid bilayer. The post-inserted ICG may be partly dissolved in the lipid bilayer, because the *T_m_* values of liposomes with ICG were slightly lower than the *T_m_* values of liposomes without ICG (42.6–42.7 °C and 41.9–42.3 °C for liposomes without and with ICG, respectively, Table 2). Based on molecular modeling simulations [16,31], ICG molecules can also bind to the PEG chains on the surface of the lipid bilayers. Thus, ICG is likely to be partly solubilized in the lipid bilayer and interacts with its more polar regions with surface-grafted PEG.

Most studies on liposomal oligonucleotide delivery systems have utilized either cationic or ionizable lipids. We chose non-cationic lipids for two reasons. Firstly, we wanted to specifically investigate the light-activated delivery. Since cationic as well as ionizable lipids were known to interact with cell membranes and in this way induced intracellular delivery and release [32,33,34], we preferred to study the light-activated process with neutral lipids and exclude the possible effects of charged lipids. Secondly, compared to cationic liposomes, neutral liposomes were less toxic, more stable, and showed improved pharmacokinetics [35,36,37]. However, neutral liposomes were generally less efficient in intracellular delivery and transfection, and their endosomal escape and intracellular release properties were poor [38,39]. We aimed to improve the drawbacks of neutral liposomes with light-induced release of oligonucleotides.

We demonstrated effective light-induced release and cell transfection with the ICG liposomes. ICG was required for the light-induced contents’ release from the liposomes (Appendix A), and the ICG dose was the major factor influencing the transfection efficacy (Figure 3A–D). The oligonucleotide and lipid concentrations (Figure 3E,F) and extent of light exposure (Figure 4) were less important. The liposomes efficiently released both the encapsulated SSO (*M_W_* ~ 7 kDa) and siRNA (*M_W_* ~ 14 kDa) upon light activation (Figure 1), suggesting that the system was functional with oligonucleotides of different sizes. Light exposure did not affect the activity of the SSO (Appendix A).

Interestingly, free ICG and SSO in solution also led to transfection upon light activation (Figure 5), indicating that the intracellular delivery of ICG and light was not dependent on the liposomal components. However, light triggering without ICG did not lead to any detectable transfection. Hence, ICG is a necessary component for the light-triggered delivery, while the liposomes are optional. The use of non-liposomal, free ICG and oligonucleotides can be applied as a simple delivery system to certain tissues. For example, in the treatment of eye diseases, ICG and oligonucleotides may be injected into the vitreous (to treat posterior eye diseases) or instilled topically (to treat corneal diseases). The small size of naked oligonucleotides and ICG compared to liposomes may be beneficial for penetration into certain targets. The combination of ICG and oligonucleotides to a liposomal formulation has, however, certain advantages. Firstly, this approach avoids problems related to differences in clearance kinetics of oligonucleotides versus ICG. Secondly, it ensures that ICG and the oligonucleotides are taken up by the same cell, which is a requirement for the light-induced transfection. Thirdly, liposomes stabilize ICG in vivo [40,41]. Fourthly, liposomes can be targeted to selected cells by attaching suitable ligands on their surface [42]. Dual targeting with ligands and light triggering could maximize the therapeutic effect in target cells and minimize the exposure of non-target cells.

In general, oligonucleotides should cross at least two cellular membranes to reach their sites of action inside cells in the cytosol or in the nucleus: firstly, they should permeate the cellular membrane surrounding the cell (cellular uptake) and secondly, the endosomal membrane after cellular uptake (endosomal escape). As large and anionic molecules, naked oligonucleotides are not easily taken up by negatively charged cells. Further, the endocytosed oligonucleotides are not released from endosomes, but transferred to lysosomes, where they are degraded enzymatically [43,44,45]. Cationic and ionizable lipids interact with negatively charged cellular membranes inducing cellular uptake and endosomal escape [32,46]. With ICG and light triggering, the mechanism must be different, as transfection takes place without cationic lipids or even without any delivered lipids (free ICG, Figure 5). A possible mechanism for the increased cellular uptake and endosomal escape induced by ICG and light is partial fluidization of the cellular membranes, allowing oligonucleotides to permeate and cross the cellular lipid bilayers. This hypothesis is supported by the ability of free ICG without lipids to induce transfection, while in the absence of ICG, no transfection takes place (Figure 5B).

We suggest that the increased fluidity of cell membranes and consequent increase in cellular permeability by this delivery system is dependent on two factors: Firstly, the ability of ICG to diffuse into cellular membranes, and this way, to create leakiness. Secondly, released heat of ICG after light exposure causes changes in the organization of lipid bilayers, leading to fluidization of the membranes. The first factor is supported by the ability of free ICG and ICG liposomes to induce some transfection in the absence of light (20% transfection efficacy without light exposure in Figure 3D,F). Yet, the major effect on cellular membrane permeability and successful transfection is most likely the effect of increased temperature after light triggering. We have previously shown using gold nanoparticle-containing liposomes that light activation increases endosomal escape and intracellular release [47]. It is, therefore, likely that a similar mechanism is involved in the endosomal release of oligonucleotides with ICG liposomes and light.

The light exposure or liposome administration separately did not affect cell viability (Figure 7A,B). However, the combination of these treatments reduced cell numbers indicating cytotoxicity (Figure 7C). In general, the observed toxic effects were mostly dependent on the ICG dose. Yet, the effects on cell viability were transient, as the cells reverted to normal state in 4 days (Figure 7C). Apparently, the transfection process causes stress for the cells, leading to slower growth and decreased viability for a certain period. Importantly, the percentage of transfected cells also increased over time (Figure 6). This proves that the observed increase in cell numbers was not solely dependent on growth of the cells that were non-transfected, but also on the transfected cells as they remained viable. A likely mechanism for the cytotoxicity is the local heat production by ICG after light exposure, causing toxic hyperthermia for the cells. However, no increased temperature in the medium surrounding the cells could be detected after treatment with ICG liposomes and light triggering (data not shown). We assume that the quick and transient increase in temperature produced by ICG [18] leads to highly localized temperature effects not detectable in the bulk surroundings.

As such, this light-activated delivery system is potential for the local delivery of oligonucleotides into tissues that can be exposed to light, including the eye, skin, lungs, the gastrointestinal tract, and tumours. The system is excited with NIR light of 800 nm that penetrates deep into tissues [28]. Moreover, utilization of fiberoptic technologies enables light exposure of tissues that cannot be reached by superficial illumination. For example, the posterior segment of the eye is an interesting target. Retinal disorders are the leading causes of impaired vision and blindness, but drug delivery to this tissue is challenging [48,49]. Since transparent ocular tissues allow straightforward light exposure of the posterior eye and neutral nanoparticles diffuse in the vitreous without aggregation, the light-activated liposomes might be suitable for the delivery of therapeutic oligonucleotides in the posterior eye tissues via intravitreal injection [17,50,51]. Another potential application is cancer therapy, where the site- and time-controlled delivery could be used to reduce the off-target effects of toxic anti-cancer drugs. The treatments with ICG-containing liposomes can be further optimized by utilizing the detection of ICG fluorescence in vivo, allowing imaging-guided drug delivery.

## 5. Conclusions

The present study describes a light-induced method for the cellular delivery of oligonucleotides. The light-triggered ICG liposomes released and delivered oligonucleotides into cultured cells. Interestingly, free ICG also facilitated oligonucleotide delivery to cells in the presence of low intensity NIR light. This method of oligonucleotide delivery avoids the need for cationic lipids that are often accompanied with the activation of innate immune pathways, and at high doses even hepatotoxicity. As such, the light-triggered liposomes might be a good system for local delivery of therapeutic oligonucleotides at places where light penetration is not limited.

## Figures and Tables

**Figure 1 pharmaceutics-11-00090-f001:**
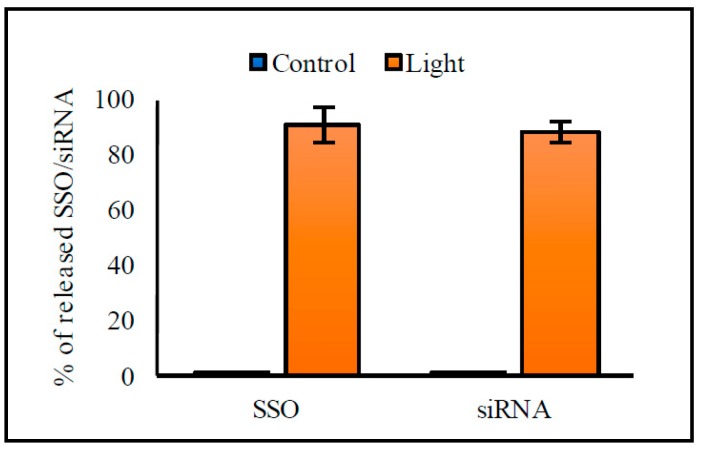
Light-triggered release of splice switching antisense oligonucleotide (SSO) and siRNA from the indocyanine green (ICG) liposomes. The liposomes were kept at 37 °C and exposed to 808 nm light with the intensity of 370 mW/cm^2^ for 1 min (light-triggered samples, orange columns) or shielded from the light (control samples, blue columns). The columns represent average values of released SSO or siRNA (*n* = 3) with error bars as standard deviation.

**Figure 2 pharmaceutics-11-00090-f002:**
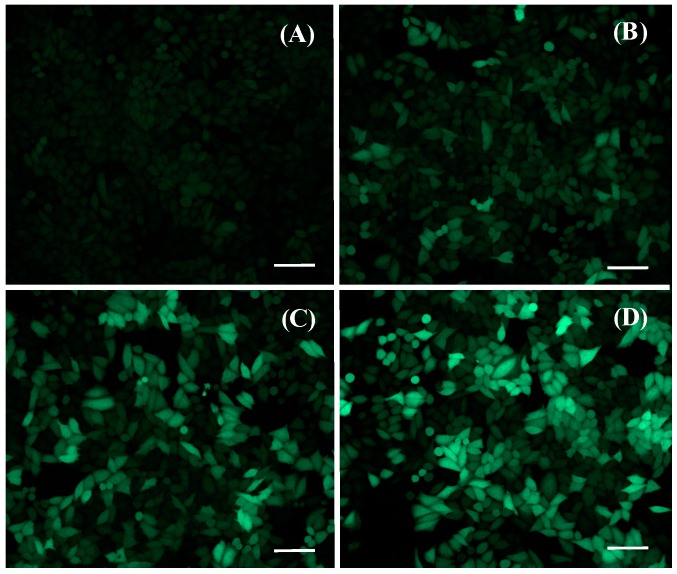
Confocal images of HeLa S3 IVS2-654 enhanced green fluorescent protein (EGFP) cells. Action of SSO restores EGFP expression. The cells were administered with (**A**) 0, (**B**) 0.5, (**C**) 0.7, and (**D**) 1.4 mM of SSO-encapsulated ICG liposomes and exposed to 808 nm light with the intensity of 370 mW/cm^2^ for 2 min. The green color represents EGFP fluorescence. The concentrations refer to lipid concentrations of the liposome dispersions. The liposomes contained ICG at a molar ratio of 1/50 ICG to lipid. Scale bar = 100 μm.

**Figure 3 pharmaceutics-11-00090-f003:**
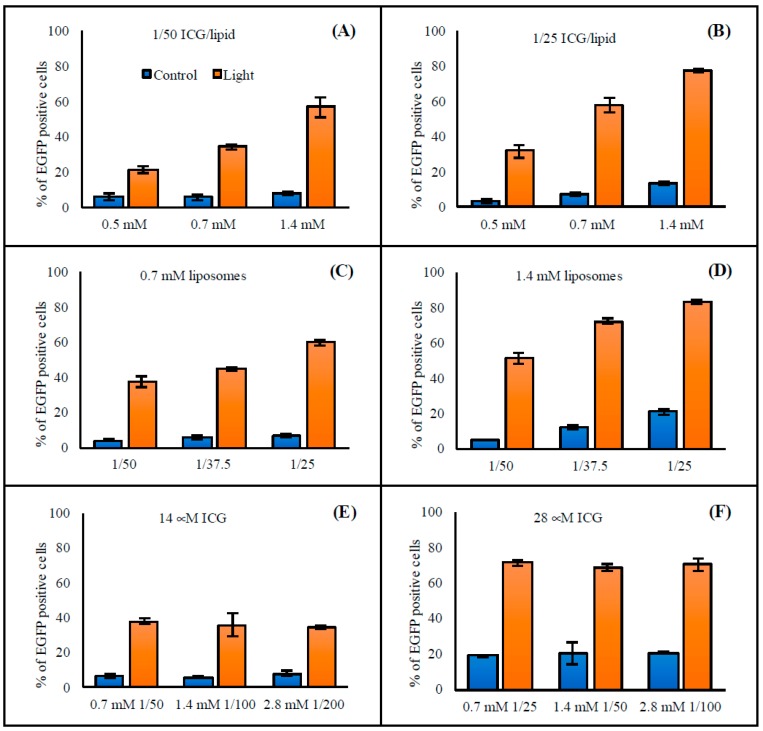
Effect of (**A**,**B**) liposome concentration and (**C**–**F**) ICG concentration of the liposomes on the transfection efficacy. The HeLa S3 IVS2-654 EGFP cells were incubated with liposomes having the following lipid concentrations and ICG-to-lipid molar ratios: (**A**) 0.5 mM–1.4 mM, 1/50 ICG/lipid, (**B**) 0.5 mM–1.4 mM, 1/25 ICG/lipid, (**C**) 0.7 mM, 1/25–1/50 ICG/lipid, (**D**) 1.4 mM, 1/25–1/50 ICG/lipid, (**E**) 0.7 mM–2.8 mM, 1/50–1/200 ICG/lipid, and (**F**) 0.7 mM–2.8 mM, 1/25–1/100 ICG/lipid. In (**E**) and (**F**), all the treatments had the same ICG concentration, 14 μM in (**E**) and 28 μM in (**F**). After incubation, the cells were exposed to 808 nm light (370 mW/cm^2^ for 2 min). The columns represent average values of EGFP-positive cells (*n* = 3) with error bars as standard deviation. Cells without treatment had less than 1% of EGFP-positive cells.

**Figure 4 pharmaceutics-11-00090-f004:**
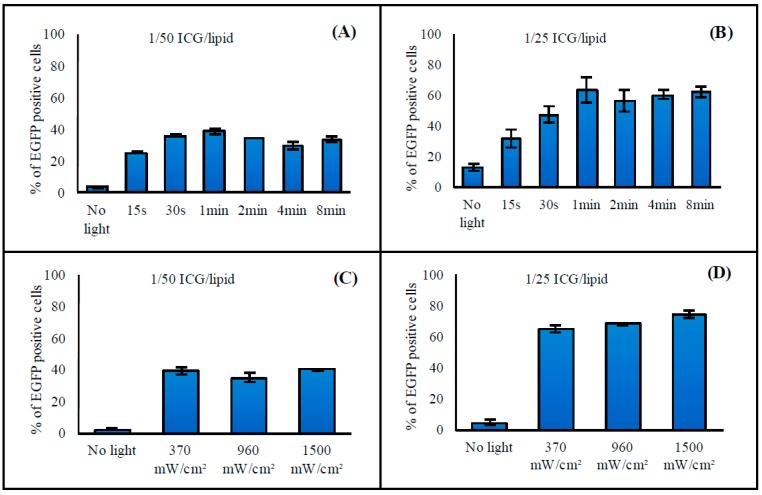
Effect of (**A**,**B**) illumination time and (**C**,**D**) light intensity on transfection efficacy. The HeLa S3 IVS2-654 EGFP cells were incubated with 0.7 mM liposomes having (**A**,**C**) 1/50 and (**B**,**D**) 1/25 ICG-to-lipid molar ratio. After incubation, the cells were exposed to 808 nm light with (**A**,**B**) the intensity of 370 mW/cm^2^ for 15 s to 8 min or (**C,D**) the intensities of 370–1500 mW/cm^2^ for 2 min. The columns represent average values of EGFP-positive cells (*n* = 3) with error bars as standard deviation. Cells without treatment had less than 1% of EGFP-positive cells.

**Figure 5 pharmaceutics-11-00090-f005:**
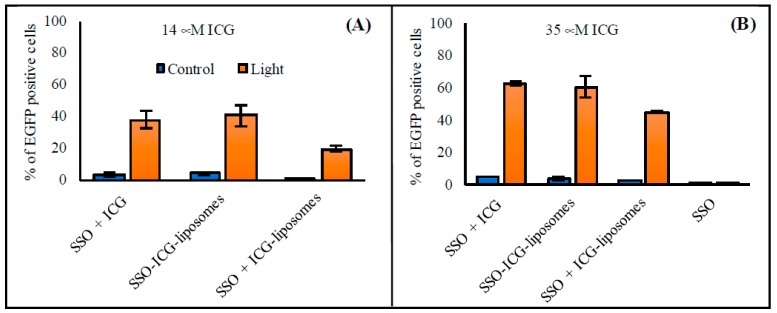
Free ICG- and SSO-induced transfection after light exposure. The HeLa S3 IVS2-654 EGFP cells were incubated with free ICG or empty ICG liposomes and SSO corresponding to the following concentrations of SSO-encapsulated ICG liposomes: (**A**) 0.7 mM, 1/50 ICG-to-lipid ratio (240 nM SSO, 14 μM ICG) and (**B**) 1.4 mM, 1/40 ICG-to-lipid ratio (480 nM SSO, 35 μM ICG). After incubation, the cells were exposed to 808 nm light (370 mW/cm^2^ for 2 min). The columns represent average values of EGFP-positive cells (*n* = 3) with error bars as standard deviation. Cells without treatment had less than 1% of EGFP-positive cells.

**Figure 6 pharmaceutics-11-00090-f006:**
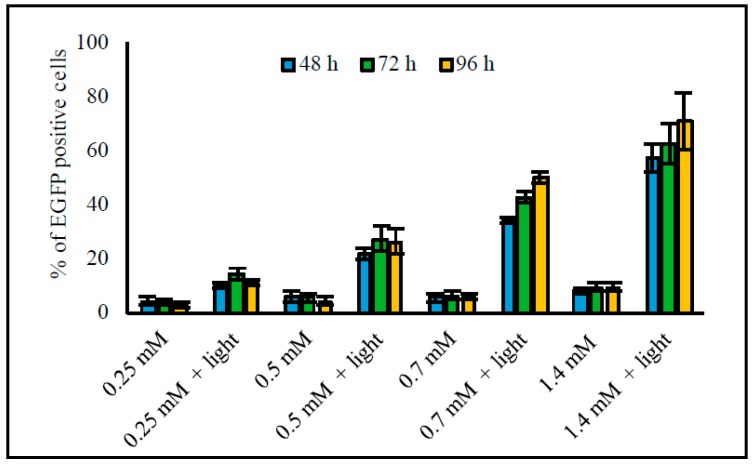
Effect of time after transfection and light triggering on transfection efficacy. The HeLa S3 IVS2-654 EGFP cells were incubated with 4 different liposome concentrations having 1/50 ICG-to-lipid molar ratio. Light exposure was performed at 808 nm (370 mW/cm^2^ for 2 min), and transfection efficacy was measured at 48, 72, and 96 h after light triggering. The columns represent average values of EGFP-positive cells (*n* = 3) with error bars as standard deviation. Cells without treatment had less than 1% of EGFP-positive cells.

**Figure 7 pharmaceutics-11-00090-f007:**
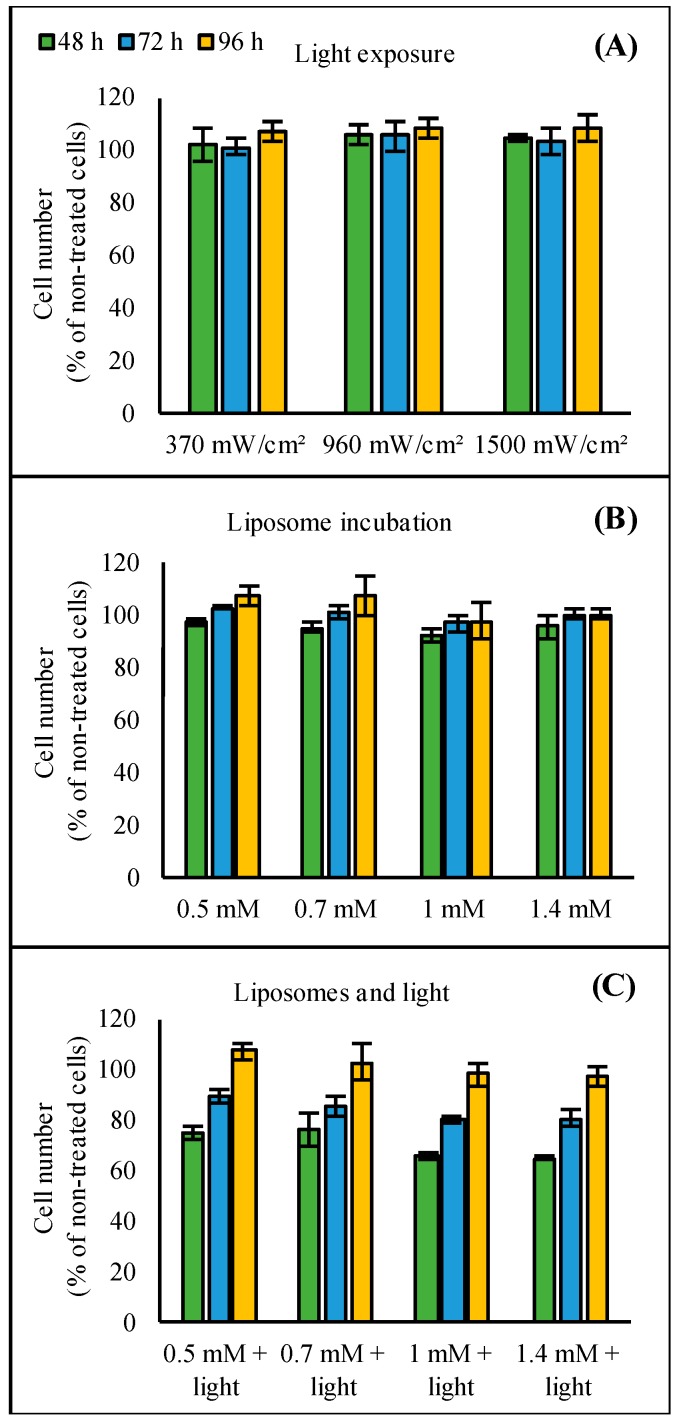
Effect of (**A**) light exposure, (**B**) the ICG liposome concentration, or (**C**) light exposure and the ICG liposome concentration on cell growth. Cells were (**A**) exposed to 808 nm light with the intensities of 370–1500 mW/cm^2^ for 2 min or (**B**) incubated with 0.5–1.4 mM liposomes having 1/50 ICG-to-lipid molar ratio overnight. In (**C**), light exposure was performed after overnight liposome incubation. The columns represent average values of cell numbers (*n* = 3) calculated as percentage of non-treated control cells with error bars as standard deviation.

**Table 1 pharmaceutics-11-00090-t001:** Size, PDI, and encapsulation efficiency of the liposomes. SD, standard deviation (n = 3) and PDI, polydispersity index.

Liposome Type	ICG Concentration	Diameter ± SD (nm)	PDI	Encapsulation % ± SD
SSO liposomes	without ICG	145 ± 3	0.123	6.7 ± 0.9
	with 1/25 ICG/lipid	150 ± 1	0.102	
siRNA liposomes	without ICG	153 ± 2	0.141	5.8 ± 0.7
	with 1/25 ICG/lipid	155 ± 2	0.116	

**Table 2 pharmaceutics-11-00090-t002:** Zeta-potential and transition temperature (T_m_) of the liposomes. SD, standard deviation (*n* = 3).

Liposome Type	ICG Concentration	Zeta-Potential ± SD (mV)	*T_m_* (°C)
SSO liposomes	without ICG	−8.1 ± 0.1	42.7
	with 1/50 ICG/lipid	−12.9 ± 0.3	42.1
	with 1/25 ICG/lipid	−16.4 ± 0.3	41.9
siRNA liposomes	without ICG	−8.1 ± 0.3	42.6
	with 1/50 ICG/lipid	−12.2 ± 0.3	42.0
	with 1/25 ICG/lipid	−16.5 ± 0.3	42.1
Empty liposomes	without ICG	−8.2 ± 0.1	42.7
	with 1/50 ICG/lipid	−11.9 ± 0.4	42.3
	with 1/25 ICG/lipid	−15.9 ± 0.2	42.2

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
