# Peer review of "Light-Triggered Cellular Delivery of Oligonucleotides"

_pharmaceutics, 2019, doi:10.3390/pharmaceutics11020090_

Round 1
Reviewer 1 Report
In this manuscript, Kontturi et al. reported a light-sensitive carrier for oligonucleotide delivery. The carrier is a liposome that encapsulates ICG as the photo-sensitizer and oligonucleotide as the cargo. Upon excitation by 808 nm light, ICG produces heat to break down liposome and release encapsulated oligonucleotide. This work is interesting and I would like to recommend its publication after addressing following issues:
1. Is the encapsulated oligonucleotide stable enough when exposure the light irradiation or heat generated by ICG?
2. The liposome with ICG and oligonucleotide is negatively charged. How could it cross negatively charged cellular membrane? The authors may give an explanation or discussion.
3. Negative control oligonucleotide with scramble sequence is needed to prove the induction of gene expression is by the delivered oligonucleotide.
4. The authors also showed that even without liposome, free ICG and oligonucleotide could also enter cell after light irradiation. Please give an explanation for the reason. Is it possible oligonucleotide itself could enter cells with light irradiation?
Author Response
See attached document

Reviewer 2 Report
This study presents results on a light-triggered liposomal drug delivery system, where the authors present in vitro results on using these liposomes for cellular delivery of oligonucleotides. The paper is overall well written, the topic is relevant to the research of Pharmaceutics, and there is need for better delivery systems for oligonucleotides. I found two major issues that the authors should address:
1) The results indicate that light+ICG induced transfection, and that light+TLS also resulted in cytotoxicity. However, the mechanism for these findings was not clarified. One likely mechanism is that the ICG absorbs the light and produces hyperthermia (since this was the mechansim on which these liposomes were based). Hyperthermia is known to both be cytotoxic above certain temperatures, and to enhance membrane permeability, and could explain these effects. Thus, it would be relevant for the authors to confirm any heating that was induced during their experiment. This could be done by small temperature sensors or placed in their well plates, or infrared cameras.
2) For these liposomes to be translatable, the liposomes require adequate plasma stability. After intravenous injection, liposomes have to extravasate into tissue and then be taken up by cells. This typically requires days. I am not aware of any light- or thermosensitive liposomes with such long plasma stability. Thus, the authors either need to provide data demonstrating stability in human or animal plasma, find another workaround, or discuss this limitation in detail.
Minor issue:
Fig 1,3,4,5: should be color bars (as indicated in Fig 1 caption)
Author Response
See attached document

Round 2
Reviewer 1 Report
I am satisfied with the replies or changes made by the authors, no more revision is needed.
Reviewer 2 Report
The authors addressed all prior concerns of this reviewer.